# Impact of Five Types of Front-of-Package Nutrition Labels on Consumer Behavior among Young Adults: A Systematic Review

**DOI:** 10.3390/nu16172819

**Published:** 2024-08-23

**Authors:** Zhiyi Guo, Yueyue Ning, Muhizam Mustafa

**Affiliations:** School of the Arts, University Sains Malaysia, Penang 11800, Malaysia; zhiyi@student.usm.my (Z.G.); ningyueyue@student.usm.my (Y.N.)

**Keywords:** front-of-package nutrition labels, consumer behavior, purchase behavior, purchase intention, young consumer

## Abstract

The World Health Organization (WHO) recommends that governments promote and implement front-of-package (FoP) nutrition labels to prevent noncommunicable diseases. Recently, additional research has focused on various views on FoP label creation. However, few review studies have examined how different FoP labels affect young consumer behavior. Therefore, this study thoroughly analyzed the evidence on how FoP labels in five categories affect young consumer purchasing intention and behavior. We searched for keywords in the Web of Science, Scopus, and EBSCO databases and screened study samples according to inclusion and exclusion criteria. Fourteen studies that matched the criteria were included in this review. We discovered that numerous studies support the efficacy of the “graded indicators” category labels, with the “color-coded” and “positive logos” categories trailing closely behind. The effectiveness of the “warning” FoP label category is mixed on consumers’ healthy purchasing behavior and intention. The numerical FoP labels were most commonly used to indicate ineffectiveness. Future studies should investigate the effects of FoP labels on various subpopulations and conduct thorough evaluations of the design elements of FoP labels. Also, they should offer evidence-based recommendations, supported by both quantitative and qualitative data, for regions that have not yet implemented FoP labeling systems.

## 1. Introduction

The rates of overweight and obesity are rising sharply in a large number of countries globally, and the rising trend in overweight and obesity rates is already occurring in younger populations, which will lead to an increasing prevalence of type 2 diabetes and other chronic diseases [1,2]. This also represents a serious clinical and financial challenge for governments. Thus, the prevention of diet-related diseases is imperative. Front-of-package (FoP) nutritional labels are a supplemental nutritional information to help consumers understand the nutritional value of products, thus increasing consumer choices of healthier food products in the same category [3]. So much so that the World Health Organization (WHO) encourages more governments to promote and implement FoP labeling strategies to prevent noncommunicable diseases [4]. Effective FoP nutrition labeling strategies have significant benefits from a variety of perspectives, with Gokani & Garde (2023) concluding that consumers are better informed about the nutrient content of food. Economic operators enjoy a fair competition environment and legal protection. Governments can more effectively develop policies to reduce obesity and diet-related diseases and deliver on their international commitments to promote healthier food environments, and they can facilitate the operation of the internal market through a harmonized FoP labeling scheme to ensure a high level of consumer and health protection [5].

The increasing abundance of processed foods in the market and the global increase in obesity and overweight patients has led to a wide variety of FoP labels in different formats [6]. FoP label types are classified in typologies with two main characteristics [7]. One is related to the level of “directiveness” of the labeling system, meaning the extent to which the label design provides consumers with an interpretative direct indication of the nutritional value of the food product [8]. The first has been classified by scholars into two categories based on whether or not they provide an interpretation of the nutritional information, reductive labels and interpretative labels (Figure 1), where reductive labels simplify the impact information in a nutrition facts panel (NFP) but do not provide any interpretation of this information [9], while interpretive labels provide a further assessment of the information in the NFP [10,11].

Another classification method is the “nutrient-special” scheme, which is used in this article and offers detailed nutritional data for particular nutrients, as well as the “summary indicator” scheme, which offers a thorough examination and assessment of the food product’s overall nutritional value (Figure 2) [12]. The “nutrient-special” category can be further separated into the subcategories “numerical”, “color-coded”, and “warning labels”. Numerical category labels indicate the energy and key nutrient information per serving on the front of the package [13]. Another nutrient-special labeling system with color-coded categories, as opposed to monochromatic numerical category labels, the format of classic traffic-light labels (TLLs) uses red, orange, and green colors to indicate whether a food has high, medium or low levels of saturated fats, sugars, and sodium [14]. In recent years, warning labels in the nutrient-special category have attracted much attention. Warning labels usually appear with simple words such as “high in sugar” and “high in sodium” [15]. The summary label type includes the subcategories “graded indicators” and “positive logos” [6]. The graded indicators category of labeling is based on an impact analysis model of food standards and shows the overall nutritional quality of the food in terms of ratings [16]. Another type of simple summary label shows only one symbol, such as a tick mark or a pyramid, that is, “positive logos”, which provide a comprehensive evaluation of the overall nutritional value of the product [17]. In summary, FoP labeling systems currently on the market can be classified into five categories: “numerical”, “color-coded”, “warning labels”, “graded indicators” and “positive logos”.

Consumer behavior has been clearly defined by several scholars. Bergadaà and Faure (1995) consider consumer behavior as how and why people consume products [18]. Zikmund and D’Amico (1996) mentioned that consumer behavior corresponds to the activities of people engaging in choices and purchases [19]. Agarwala et al. (2019) stated that consumer behavior refers to the attitudes, values, and behaviors that are exposed from the perspective of consumption [20]. Nassè (2021) summarized that consumer behavior refers to the mode of buying or repurchasing guided by certain criteria such as choice, consumption, quality, taste, advertising, or price expectations. [21]. The study on consumer behavior is crucial as it helps companies, marketers, and researchers to identify and understand the various elements of the consumer decision-making process in order to enhance customer experience and promote product innovation [22]. However, the main purpose of FoP labeling is to quickly inform consumers about the relative healthiness of a product in order to help their decision to make healthier choices [23]. This research primarily examines consumer conduct in purchasing decisions, specifically the influence of FoP labels on consumer purchase behavior and intentions. However, it does not consider changes in customer attitudes toward FoP labels or their objective understanding of food goods.

An increasing amount of research has concentrated on different perspectives about the development of FoP labels, and there are also specific challenges in FoP labeling strategies. Ikonen et al. (2019) pointed out that some types of FoP labeling systems have been criticized, and different types of FoP labels have different effects on guiding consumers to make healthier choices [10]. However, the FoP labeling categories summarized in this article only include interpretative labels and reductive labels, or summary labels and nutrient-specific labels, and do not mention the specific five label categories in detail. This may cause some confusion for some scholars who are new to this field. An et al. (2021) also mentioned in a review study that research results remain mixed and inconclusive regarding the effectiveness of FoP label design in driving consumers to purchase healthier foods [24]. Moreover, Braesco and Drewnowski (2023) highlighted that FoP labels have the potential to enhance consumer behavior and the nutritional value of packaged foods through the examination of four different FoP label designs [25]. However, the existing evidence is inadequate, and further research is necessary to enhance the features and usage conditions of FoP labels. Nevertheless, some literature points out that the differences in the effectiveness of FoP labels may be due to the different effects of nutrition labels on population subgroups [26,27]. Therefore, combined with the background of younger people with diet-related diseases, the main purpose of this study is to summarize the findings of existing FoP labeling research on the impact of young people’s consumer behavior to evaluate the effectiveness of the five categories of FoP labeling systems for young subgroups. The insights gained from this study are the basis for a deeper understanding of FoP nutrition labeling, which will build on the gaps in review articles summarizing FoP labeling performance in young subgroups and will be of great significance to scholars who pay attention to and understand the related fields and governments and institutions who attempt to develop new FoP labeling design models.

## 2. Materials and Methods

### 2.1. Study Selection Criteria

This review included studies that satisfied all of the following criteria: (1) the study focused on the topic of FoP nutrition labels. (2) the outcome of the study was the effect of FoP labels on consumer behavior, including purchase behavior and purchase intention. (3) the study design was a randomized controlled trial, cohort study, pre–post study, or cross-sectional study. (4) the population was young people aged 18 to 30 years, and college students were included. (5) the type of article was a peer-reviewed publication. (6) the search period was from the date of the creation of the database to 15 May 2024. (7) the language was English.

This review excluded studies that met any of the following criteria: (1) case reports or case–control studies. (2) studies that examined other types of food labels rather than FoP labels. (3) non-English-language studies. (4) non-peer-reviewed articles. (5) studies in which the FoP labels were not actual labels in the real world but only hypothetical label designs.

### 2.2. Search Strategy

On 15 May 2024, a keyword search was conducted in three databases, Web of Science, Scopus, and EBSCO. The keywords used in the search strategy are shown in Table 1. The titles and abstracts of the articles identified through the keyword search were screened according to the study selection criteria. Then, we conducted a full text evaluation of potentially relevant articles.

### 2.3. Data Extraction and Synthesis

The screening process and results are shown in Figure 3. The initial search produced a total of 1598 results. After removing duplicate entries, 947 articles remained. During the initial screening phase, two reviewers (G and N) read the titles and abstracts of the articles according to the inclusion and exclusion criteria. Of these 947 articles, reviewer G marked 84 articles in the initial round of screening, while reviewer N marked 90 articles. Of these, 83 articles were repeatedly marked by them. After identifying articles with conflicts between the two reviewers, the two reviewers finally identified 86 articles to be marked. They required more in-depth analyses to determine whether they met the criteria. During the second round of screening, we thoroughly searched the complete text of studies that had the potential to meet the criteria, utilizing library and web resources. Two reviewers (G and N) thoroughly examined the complete content of 86 articles. After a thorough evaluation, 29 articles were excluded because they did not answer the question of the impact of FoP labels on consumer behavior, 27 articles were excluded because there were no young people of age 18–30 years old in the participant population or young people were not analyzed, 16 articles were excluded because they were not real FoP labels but hypothetical labels, and finally, 14 articles were selected for inclusion. The agreement between the two reviewers was almost perfect as shown by Cohen’s kappa measurement (κ = 0.83) [28].

All literature and databases were kept in Endnote 21. The review covered several study approaches and a diverse set of outcome measures, making it impractical to undertake a meta-analysis. Therefore, a narrative synthesis of these studies was performed. The studies were categorized based on the five types of FoP labels and subsequently investigated the influence of various label formats on consumer behavior.

### 2.4. Study Quality Assessment

Each article included in this review underwent a quality assessment based on eight criteria derived from the standards for literature evaluation provided by Kitchenham & Charters (2007) [30], including the following: (1) Were the aims of the study clear? (2) Was the study specifically designed to address those objectives? (3) Did the study have a control group? (4) Was the study sample size reasonable? (5) Were the data collection procedures adequately clear? (6) Was the study able to provide an explanation on the reliability and validity of the measurements? (7) Do the results add to the literature? (8) Does the study add to your knowledge and understanding? The scoring of each question was based on a three-point scale, where “yes” was assigned a score of 1, “no” was assigned a score of 0, and “partial” was assigned a score of 0.5. Hence, the overall score for study quality varied between 0 and 8. Study quality scores can be used to assess the strength of the evidence for a study [31], but they were not employed to judge whether a study should be included. A higher total score represents the degree to which the study addresses the research question.

### 2.5. Coding

The initial coding was first carried out independently by reviewer G, and then another reviewer, N, checked all data extractions. The initial coding table included the study ID, authors, year of publication, journal of publication, country, FoP label format, FoP label type, study methodology, study design, food product object of study, sample of study, and main findings. The main findings box was mainly filled with the conclusions of the selected articles that answered the research questions of the study. After finishing the initial coding, the matrix was managed according to the different FoP label categories.

## 3. Results

There were a total of 14 papers in the final review and analysis [16,32,33,34,35,36,37,38,39,40,41,42,43,44]. They were all published between 2015 and 2023. As in Figure 4, three of them were cross-country studies, and the rest were conducted in France (n = 4), Australia (n = 1), New Zealand (n = 1), Switzerland (n = 1), Italy (n = 1), Uruguay (n = 1), Jamaica (n = 1), and India (n = 1). The study designs consisted of a randomized controlled trial (RCT) (n = 12) and cohort study (n = 2). Three of the articles were studies of college students, one was of 18–35 year olds, and the remaining nine studies analyzed sociodemographics, with subgroups including 18–30 year olds. All of these studies were quantitative in nature, and their study sample sizes ranged from 100 to 21,702 individuals.

Studies of the five types of FoP labels were marked to be tabulated, as shown in Table 2. Some studies looked at how different types of FoP labels affected how people behaved, so the study IDs may show up more than once in the grid of different FoP label types. The FoP label studies of the numerical category have the study IDs 01, 02, 03, 05, 06, 08, 09, 12, and 14, totaling nine [16,32,33,34,36,37,39,42,44]. The color-coded category has the study IDs 01, 02, 03, 06, 08, 09, 12, 13, and 14, for a total of nine [16,32,33,34,37,39,42,43,44]. Studies on the FoP labels of the warning labels type have the study IDs 04, 06, 08, 09, 11, 12, and 13, for a total of seven [16,35,37,39,41,42,43]. FoP label studies on the positive logos category have the study IDs 01 and 14, for a total of two. FoP label studies on the graded indicators type have study IDs 01, 03, 04, 05, 06, 07, 08, 09, 10, and 12, for a total of ten [16,32,34,35,36,37,38,39,40,42].

Two of the fourteen studies were not set up to study food product objects. Breakfast cereal was the food product most included in the study (n = 8), followed by pizza (n = 7), and then cake (n = 6). The food product object least included in the study was sugar-sweetened beverages (SSBs) (n = 1).

Table 3 reports the results of the study quality assessment for specific criteria. The 14 included studies rated in the range of 5–8 (out of a total score of 8), with an average score of 7 for the studies. All of these included studies had a clear research aim and a clear study design, but only four of the included studies had a detailed explanation of the origin of the sample size. The studies conducted by Fuchs et al. (2022) and Hamlin et al. (2015) mention the number of randomly recruited participants but do not provide details on the recruitment process or an explanation for the selection criteria [33,40]. In quantitative research, however, the control of sample size is a very important part of the process. The approach to data collection was clear and unambiguous in all the studies, and the validity of the measurements was also explained. The overall focus on younger consumers was of average relevance across the 14 studies. Because several of the studies did not have significant differences in the results across sociodemographics, the authors did not discuss individual subgroups. Combining the sample sizes with the discussion of the authors in the articles, the two studies that focused primarily on college student samples and the two studies that analyzed sociodemographics had the highest relevance [16,32,33,35].

### 3.1. Numerical Labels

Table 4 is based on the initial coding omitting journals, geography, methodology, study design, study food product objects, and study samples, and the convergence of the main findings to the main findings on numerical labels. In total, nine studies reported the impact of the numerical type of FoP labels on young consumers’ purchasing behavior and purchase intention. One study suggested that the numerical category of the FoP labels had a big, positive impact on the intention of a sample of college consumers in purchasing cereal products [33]. Another cross-national study in the UK, Germany, Poland, and Turkey noted that all four FoP label formats in the study performed similarly, and all guided consumers to healthier food choices when purchasing pizza, yoghurt, and biscuits, with the study containing the numerical, color-coded, and positive logos categories [44]. Four other studies mentioned the very low effectiveness of numerical category FoP labels on consumers’ choices of healthier products [16,34,35,36]. This could be because numerical-type FoP labels require a lot of complex mental processing, which could make them less useful in consumer settings [32,34]. Unfortunately, this just means that the labels are hard to understand. The remaining three RCTs found that numerical category FoP labels have difficulty influencing most consumers to purchase healthier food products [32,39,44].

### 3.2. Color-Coded Labels

Table 5 summarizes the nine studies on color-coded category labels exploring the impact of FoP labels in this category on young consumers’ food choices, purchases, and intentions. Of these, five RCT studies and one cohort study indicated that MTL was effective in helping consumers choose healthier foods [16,32,33,34,36,44], but three of these studies showed that the Nutri-Score label format worked better than the MTL format [16,32,36]. Julia et al. (2017) noted that a subgroup of young people with a college education and lower income perceived the MTL format best, including identifying and choosing healthier products [34]. This cohort study also mentions that the attractiveness of MTLs is likely to be related to the color profile of such labels, as color is thought to contribute to the salience of FoP labels in general, whereas multiple numeric messages are often considered difficult to understand. The remaining three articles report that MTL is ineffective in helping consumers make healthier food choices [39,42,43]. Even White-Barrow et al. (2023) mentioned that MTL is ineffective in helping consumers to correctly identify the least harmful products and choose to buy the least harmful products or none at all [43]. Moreover, Singh et al. (2022) argued that although there is no evidence that this label can change consumer shopping behavior, it can improve the ability of participants to correctly identify the content of nutrients of concern such as sugar, saturated fatty acids, and sodium in packaged foods and beverages [42].

### 3.3. Warning Labels

Table 6 demonstrates the key findings on purchase choice and purchase intention for the seven studies that focused on the warning labels category. There were six RCT studies and one cohort study, and all six RCT studies used a virtual shopping design. A study on the impact of the FoP labeling of SSBs on beverage choice, health knowledge, and awareness noted that warning labels were effective in reducing the choice of SSBs in online scenarios and were the most effective in comparison to HSR [35]. There is also a study that shows that the octagonal warning label (OWL), significantly in MTL systems, helped Jamaican consumers to choose to buy the least harmful product or not to buy any product, regardless of the age, gender, and education of the population [43]. The other two studies indicated that warning labels can help consumers make healthier food choices, but the performance was the worst among the non-interpretive labels, in other words, the effect was only better than that of the numerical category labels [16,36]. Moreover, the study by Ares et al. (2018) reported that the vast majority of participants indicated that they would consider warning labels when purchasing food, but younger participants were significantly less willing to consider warnings when choosing food than participants in other age groups [41]. In addition, two other studies have shown that warning labels failed to reduce consumers’ intention to purchase unhealthy packaged products obviously [39,42].

### 3.4. Positive Logos

The two RCTs in Table 7 summarize the impact of FoP labels in the positive logos category on the behaviors and intentions of young consumers in choosing healthier foods. Both of its studies used a virtual shopping research procedure design. One of the studies designed a web-based virtual supermarket that contained foods labeled with the different FoP labels being tested. Participants were asked to act like they were shopping and choose food for a week for their family. All of the foods came from real brands and goods that you can buy in France. The study looked at four key types of food: processed meats, dairy products, savory foods, and sweets for breakfast. The final results of this study concluded that tick labels were the most effective in helping consumers choose healthier food products among the 18–30-year-old subgroup, slightly higher than 5-CNL and GDA labels [32]. Another study similarly concluded that health logo labels were able to guide consumers to healthier food choices, at least in the three main food groups studied (pizza, yoghurt, and biscuits) [44].

### 3.5. Graded Indicators

Table 8 is a summary of the 10 studies involving the impact of FoP labels in the graded indicators category on consumers’ choice of healthier food products or purchase intentions. Seven of these studies looked at the Nutri-Score format for nutritional package labels, three looked at HSR labels, and one looked at the 5-CNL format. Out of the seven studies that looked at Nutri-Score, six said that it could help people make the food products they buy healthier generally [16,34,36,37,38,40]. Egnell et al. (2018) mentioned that Nutri-Score summary labels show higher effectiveness than the other four formats of FoP labels and that consumers are more likely to understand the summary indicator label format, which can limit the potential confusion associated with the interpretation of nutritional terms [16]. After two years, Nutri-Score labeling was found to be the most successful scheme by Egnell et al. (2020) based on an RCT analysis. This scheme encouraged study participants to make healthier food choices and improved their ability to distinguish between different food products within product categories based on their nutritional quality [37]. Furthermore, the HSR label belongs to the category of graded indicators in another format and was second only to the Nutri-Score label format in terms of effectiveness in this study. Furthermore, the findings on 5-CNL point to similar results across sociodemographic and economic subgroups, with hierarchical summary labeling being the most effective, which is significantly better than simple summary and nutrient-specific labeling [32]. However, a research experiment from India on the nutritional packaging labeling of unhealthy packaged foods found that HSR performed worse than all the other FOP label types tested (Warning label, MTL, GDA) (except the controls) in most outcomes and that HSR labels even had a negative impact compared to the controls in some state regions [42]. In addition, a study by Fialon et al. (2020) found that while most participants did not alter their food choices under the unlabeled or FoP-labeled conditions, Nutri-Score design labels were the most successful in assisting consumers in objectively understanding food products, with HSR labels coming in second [39].

## 4. Discussion

This study conducted a comprehensive analysis of the effects of five different types of FoP nutrition labels on the purchasing intentions and behaviors of young customers. Fourteen studies that matched the selection criteria were considered for inclusion in this review study. This study evaluated various FoP labels of different categories, including GDA and RIL in the “numerical” category; MTL and TLL in the “color-coded” category; OWL and MGG in the “warning labels” category; green tick and health logo in the “positive logos” category; and 5-CNL, Nutri-Score, SENS, and HSR in the “graded indicators” category. Regarding the influence of an FoP nutrition labeling system on consumer behavior in promoting young consumers’ purchasing behavior and purchasing intention of healthier foods, most research results show that it is effective, but there are still a few research results showing no significant effect or ineffectiveness.

Of these, the Nutri-Score scheme in the graded indicators category was considered the most effective FoP label format in the reviewed studies, followed by MTL in the color-coded category. The reason for such results is likely due to the color profile. Julia et al. (2017), Egnell et al. (2018), and Egnell et al. (2020) discuss the use of color as being essential for the saliency of FoP labels [16,34,37]. Both the Nutri-Score and MTL label formats have multicolored scales from green to red, which are easier to be interpreted and understood [45,46]. Certainly, it is not only the color in design elements that significantly affects FoP label effectiveness; FoP label size and location, graphics and icons, and informational simplicity have all been discussed by scholars as well [47,48,49]. However, color is generally considered to be the design element that most significantly affects the effectiveness of FoP labels. The two studies of the FoP labeling system in the positive logos category both noted its effectiveness in helping a subgroup of young consumers make healthier food choices. In addition, four of the seven studies on warning labels noted that they were ineffective in helping consumers make food choices, and two studies noted that they performed only better than RIL in the numerical category. However, warning labels perform well in reducing consumer purchase intentions for SSBs [35,50,51]. The presence of warning labels on beverage packaging diminishes the focus on marketing components displayed on the package and heightens the perception of the diabetes-related risks linked to sugar-sweetened beverages [52]. Ultimately, most of the findings in this analysis suggest that the numerical category of FoP label forms does not significantly influence the selection of healthier food options. The potential explanation for this phenomenon is that the complex cognitive processing needed to comprehend and utilize FoP labels in this particular category is believed to impede their effectiveness in shopping settings [32]. In other words, the quantitative information provided by such labels is difficult for consumers to understand and apply [33,34].

There are still a few limitations to consider in this review. Due to our limited capability, we only considered papers published in English, which may result in biased conclusions. Because ignoring different findings that exist in articles in other languages may introduce publication bias. Second, analyses based on the Web of Science database after searching for FoP nutrition labels show that the literature in this area is large and has shown an upward trend since 2016. However, the number of studies focusing on the impact of FoP labeling on consumers in younger age groups is still limited. Third, the quantity of studies encompassed in the review examining the impact of “positive logos” and “warning labels” FoP labeling categories on the healthier food choices of young individuals was comparatively lower than the number of studies conducted on the remaining three categories. It is worthwhile to continue to explore these two types in future studies. Furthermore, this review failed to conduct a comprehensive backward and forward search of the 14 papers that were screened. This omission may have resulted in the exclusion of important material. A subsequent study that includes a forward and backward search would have provided a more thorough analysis. In addition, few studies included in the review focused on developing countries, and future studies should focus more on developing countries to promote the implementation of FoP nutrition labeling strategies to prevent diet-related diseases [53]. In addition, most of the intervention studies reviewed in this study were in virtual supermarket environments, whereas people’s purchasing decisions are partly dependent on the purchasing environment and product brand and food flavor [24,54]. Finally, the categorization of FoP labels employed in this study is not universally applicable to other classifications. For instance, the European Commission acknowledges only four types of FoP labels: graded indicators, endorsement logos, color-coded labels, and numerical labels. This discrepancy could potentially influence the findings of this study to some extent. Despite these limitations, this review provides fair insights into the impact of FoP nutrition labeling on the purchase behavior and purchase intentions of young consumers across five categories.

There are several recommendations for advancing research in this area. Firstly, although Penzavecchia et al. (2022) noted the need for further research on the impact of FoP nutrition labeling on consumer behavior in different sociodemographic groups [26], and An et al. (2021) mentioned that large and demographically representative samples should be recruited and that studies should be conducted on different subgroups of the population according to sociodemographic characteristics [24], the research in this direction is still not enough. Secondly, most of the articles discussing the design elements of the FoP label format only discuss the color and size elements, lacking an exploration of the more comprehensive design elements and the differences in how different sociodemographic groups respond to various design elements. The design elements encompass shape, size, color, font, simplicity of labeling, and location. The many sociodemographic categories encompass not only the specific youthful subgroup that is the main subject of this article, but also other age subgroups, as well as subgroups characterized by diverse economic statuses and educational levels. Subsequent investigations could potentially examine this phenomenon by employing questionnaires or experimental designs that provide adequate reliability and validity. Thirdly, the majority of the food products examined in the evaluated publications were breakfast cereals, pizza, and cakes. However, future studies should encompass a broader range of product categories for investigation, including SSBs, chip snacks, yoghurts, cookies, and so on. Fourthly, young people are the largest group consuming SSBs [55,56], but research on the impact of FoP labels on driving healthier SSB consumption choices in the young adult subgroup is still lacking. Finally, there should be more qualitative research in this area to gain a deeper insight into the impact of FoP labels on advancing consumer behavior, as well as quantitative and qualitative data to propose a guideline for regions or countries that have not yet implemented a FoP labeling strategy.

## 5. Conclusions

In summary, this study reviewed the impact of five categories of FoP labels on consumer behavior among the young subgroup. Regarding the label design of the “graded indicators” category, its effectiveness is supported by multiple studies, while the “color-coded” and “positive logos” categories follow closely behind. Nevertheless, the effectiveness of FoP labels categorized as “warning labels” in influencing healthier purchase behaviors and intentions was inconsistent, showing mixed results. Meanwhile, multiple studies show a poor performance of the “numerical” category. In general, most FoP labeling schemes are effective to some extent for young consumers, but the effectiveness varies depending on the label category and the research subjects.

While this analysis provides a comprehensive summary of the English literature from three databases, we recognize that the bias in selecting studies may impact the generalizability of the findings. Hence, it is recommended that future studies be more diverse in sample selection to improve the generalizability of the results. We recommend that future studies incorporate a meta-analysis to yield more comprehensive and precise findings. Furthermore, when performing a meta-analysis, it is imperative to employ a funnel analysis to assess and mitigate publication bias.

However, our study still provides value to the FoP labeling field, especially for policymakers, including government agencies and nutrition associations. By comprehending the influence of different types of FoP labels on the purchase behavior and intention of young customers, researchers may put up and execute more refined recommendations to enhance the actual effectiveness of FoP labels and the decision-making process of consumers.

## Figures and Tables

**Figure 1 nutrients-16-02819-f001:**
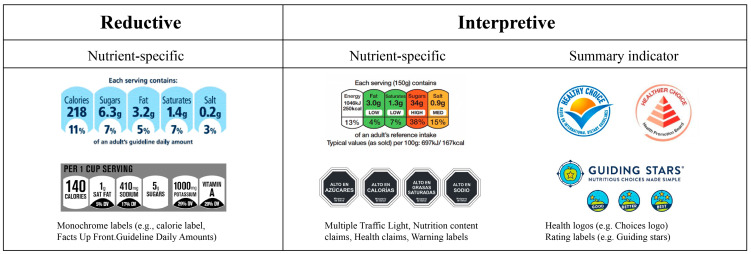
Classification of front-of-package (FoP) nutrition labels. Change according to Ikonen et al. (2019) [10].

**Figure 2 nutrients-16-02819-f002:**
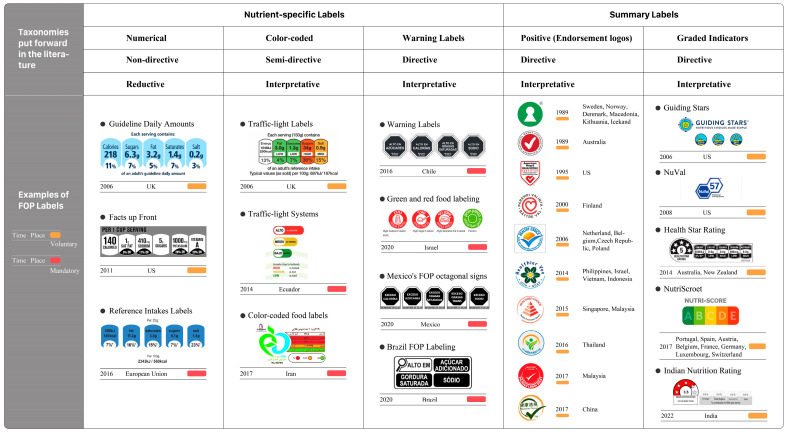
An example of the typologies and formats of FoP nutrition labeling schemes implemented/proposed/announced. Change according to the European Commission (2020) [6].

**Figure 3 nutrients-16-02819-f003:**
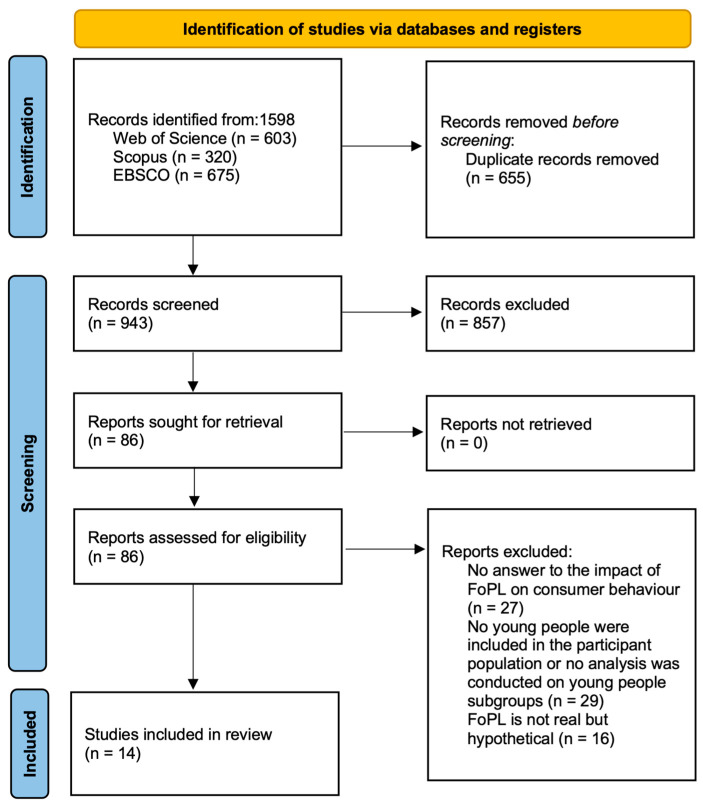
Study scanning and selection process [29].

**Figure 4 nutrients-16-02819-f004:**
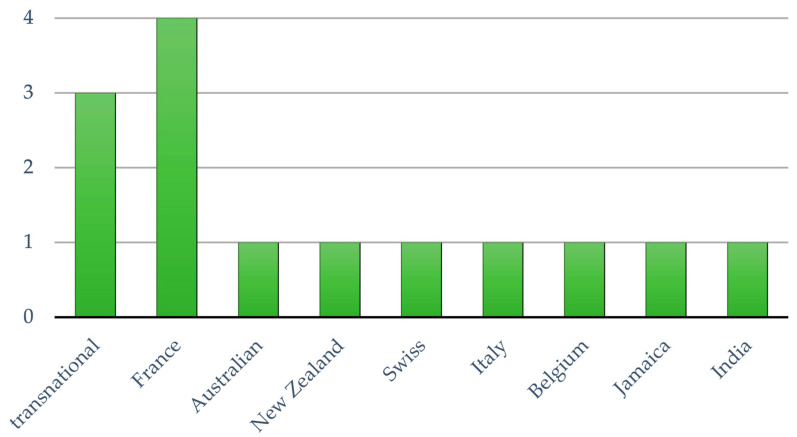
Number of selected articles in the study countries.

**Table 1 nutrients-16-02819-t001:** Search strategy keywords.

Mesh Term	Search Keywords
Front-of-package nutrition labels	“nutritional label” or “nutrition label” or “food label” or “front-of-pack label” or “front-of-package label” or “front-of-packaging label” or “nutritional packaging label” or “nutrition package label” or “Nutri-score” or “Traffic light label” or “Health star rating” or “Guideline daily amount” or “Facts up front” or “Guiding star” or “NuVal” or “numerical” or “color-coded” or “warning labels” or “positive logos” or “Keyhole” or “choice logo” or “healthier choice” or “healthy choice” or “heart symbol”
	AND
Young adult	“young adult” or “young people” or “young consumer” or “college student” or “university student” or “aged 18–30”
	AND
Consumer behavior	“consumer behavior” or “consumer behaviour” or “purchasing behavior” or “purchasing behaviour” or “purchasing intention” or “buying behavior” or “buying behaviour” or “buying intention”
	NOT
	“cigarette” or “alcohol” or “medicine” or “allergy”

**Table 2 nutrients-16-02819-t002:** Five types of FoP label study ID statistics.

Type of FoP Labels	Study ID
Numerical	01, 02, 03, 05, 06, 08, 09, 12, 14
Color-coded	01, 02, 03, 06, 08, 09, 12, 13, 14
Warning labels	04, 06, 08, 09, 11, 12, 13
Positive logos	01, 14
Graded indicators	01, 03, 04, 05, 06, 07, 08, 09, 10, 12

**Table 3 nutrients-16-02819-t003:** Quality assessment result.

Study ID	Q1	Q2	Q3	Q4	Q5	Q6	Q7	Q8	Total
01	1	1	1	1	1	1	1	1	8
02	1	1	1	0	0.5	1	1	1	6.5
03	1	1	0	0.5	1	0.5	1	0.5	5.5
04	1	1	1	1	1	1	1	0.5	7.5
05	1	1	1	1	1	1	1	0.5	7.5
06	1	1	1	0.5	1	1	1	1	7.5
07	1	1	1	1	1	1	1	0.5	7.5
08	1	1	1	0.5	1	1	1	1	7.5
09	1	1	1	0.5	1	1	1	0.5	7
10	1	1	1	0	1	1	1	0.5	6.5
11	1	1	0	0.5	1	1	1	0.5	6
12	1	1	1	0.5	1	1	1	0.5	7
13	1	1	1	0.5	1	1	1	0.5	7
14	1	1	1	0.5	1	1	0.5	0.5	6.5

**Table 4 nutrients-16-02819-t004:** Key findings of numerical category front-of-package label studies.

Study ID	(Author, Year)	Label formats	Type of FoP Labels	Key Finding about Numerical Labels
01	(Ducrot et al., 2016) [32]	GDA, 5-color Nutrition Label (5-CNL), Multiple Traffic Lights (MTL), Green Tick	Numerical, color-coded, positive logos, graded indicators	The effectiveness of GDA labeling is consistent with findings from previous studies, with no discernible effect on healthier food choices (most subgroups showed trends similar to those reported for the overall sample).
02	(Hamlin et al.,2015) [33]	MTL, RIL	Color-coded, numerical	In a group of college students, RIL, or numerical-type labels, have a statistically significant positive main effect on their desire to buy cereal goods. The efficiency is a little higher than that of labels with colors, but the difference is not very big.
03	(Julia et al., 2017) [34]	Nutri-Score, MTL, SENS, RIL	Graded indicators, color-coded, numerical	RIL is difficult to understand, and very few consumers believe that this FoP label helps them choose healthier products.
05	(Egnell et al., 2019) [36]	Nutri-Score, RIL	Graded indicators, numerical	No overall difference was observed between RIL and the unlabeled group, that is, the effectiveness of RIL on the purchase of healthier food products by student consumers was very low.
06	(Egnell et al.,2020) [37]	MTL, RIL, Warning label, Nutri-Score, HSR	Color-coded, numerical, warning labels, graded indicators	RIL helps consumers make food choices, but the effect is the smallest among the several types of FoP labels.
08	(Egnell et al.,2018) [16]	HSR, MTL, RIL, Waring label	Color-coded, numerical, warning labels, graded indicators	All interpretive labels, or in other words, numerical labels, perform significantly better than reductive labels.
09	(Fialon et al., 2020) [39]	RI, MTL, Warning label, Nutri-Score, HSR	Color-coded, numerical, warning labels, graded indicators	There are five types of FoP labels that do not help people choose healthy foods.
12	(Singh et al.,2022) [42]	Warning label, TLL, GDA, HSR	Warning labels, graded indicators, numerical, color-coded	GDA has no impact on consumers buying healthier products.
14	(Hodgkins et al., 2015) [44]	GDA, TL, MTL, Health logo	Numerical, color-coded, positive logos	All four FoP label formats performed similarly and all guided healthier food choices.

**Table 5 nutrients-16-02819-t005:** Key findings of color-coded category front-of-package label studies.

Study ID	(Author, Year)	Label Formats	Type of FoP Labels	Key Finding about Numerical Labels
01	(Ducrot et al., 2016) [32]	GDA, 5-color Nutrition Label (5-CNL), Multiple Traffic Lights (MTL), Green Tick	Numerical, color-coded, positive logos, graded indicators	MTL can help consumers choose healthier food products but not as effectively as graded indicators.
02	(Hamlin et al., 2015) [33]	MTL, RIL	Color-coded, numerical	MTL has a positive effect on consumers intentions to purchase cereal products in an experimentally controlled situation in a sample of college students.
03	(Julia et al., 2017) [34]	Nutri-Score, MTLs, SENS, RIL	Graded indicators, color-coded, numerical	MTL appears to be the second most popular FoP label in the population, particularly among college-educated young people with lower incomes.
06	(Egnell et al., 2020) [37]	MTL, RIL, Warning label, Nutri-Score, HSR	Color-coded, numerical, warning labels, graded indicators	The MTL system helps people choose healthy foods, but not as well as the Nutri-Score and HSR methods.
08	(Egnell et al., 2018) [16]	HSR, MTL, RIL, Waring label	Color-coded, numerical, warning labels, graded indicators	This is the only one besides MTL that does better than the Nutri-Score label, but the Nutri-Score has a much higher efficiency score.
09	(Fialon et al., 2020) [39]	RI, MTL, Warning label, Nutri-Score, HSR	Color-coded, numerical, warning labels, graded indicators	Five formats of FoP labels are ineffective in helping consumers make healthier food choices.
12	(Singh et al., 2022) [42]	Warning label, TLL, GDA, HSR	Warning labels, graded indicators, numerical, color-coded	There is no evidence that TLL can change consumer purchasing behavior, but it is the most informative and easy-to-understand label.
13	(White-Barrow et al., 2023) [43]	Octagonal warning label (OWL), TLL, Magnifying glass icon (MGG)	Warning labels, color-coded	TFL is less effective than warning labels in helping consumers choose healthier products. In addition, TFL is ineffective in helping consumers correctly identify the least harmful options and choose to purchase the least harmful option or none at all.
14	(Hodgkins et al., 2015) [44]	GDA, TL, MTL, Health logo	Numerical, color-coded, positive logos	All four FoP label formats performed similarly and all guided healthier food choices.

**Table 6 nutrients-16-02819-t006:** Key findings of warning labels category front-of-package label studies.

Study ID	(Author, Year)	Label Formats	Type of FoP Labels	Key Finding about Numerical Labels
04	(Billich et al., 2018) [35]	Warning label, HSR	Warning labels, graded indicators	Warning labels are effective in reducing the choice of sugar-sweetened beverages in online choice scenarios.
06	(Egnell et al., 2020) [37]	MTL, RIL, Warning label, Nutri-Score, HSR	Color-coded, numerical, warning labels, graded indicators	Warning labels help consumers’ food choices but are only better than the RIL format.
08	(Egnell et al., 2018) [16]	HSR, MTL, RIL, Waring label	Color-coded, numerical, warning labels, graded indicators	Warning labels help consumer food choices but perform the worst among non-interpretive labels.
09	(Fialon et al., 2020) [39]	RI, MTL, Warning label, Nutri-Score, HSR	Color-coded, numerical, warning labels, graded indicators	Five formats of FoP labels are ineffective in helping consumers make healthier food choices.
11	(Ares et al., 2018) [41]	Warning label	Warning labels	Younger participants aged 18–24 were significantly less willing to consider warnings when choosing food than participants in other age groups. However, the vast majority indicated that they would change their choice of products with warning labels.
12	(Singh et al., 2022) [42]	Warning label, TLL, GDA, HSR	Warning labels, graded indicators, numerical, color-coded	Warning labels fail to effectively reduce intentions to buy unhealthy packaged products.
13	(White-Barrow et al., 2023) [43]	Octagonal warning label (OWL), TLL, Magnifying glass icon (MGG)	Warning labels, color-coded	The results show that OWL significantly outperforms MGG and TLL in helping Jamaican consumers choose to purchase the least harmful option or none at all, irrespective of the age, gender, and educational level of the population.

**Table 7 nutrients-16-02819-t007:** Key findings of positive logos category front-of-package label studies.

Study ID	(Author, Year)	Label Formats	Type of FoP Labels	Key Finding about Numerical Labels
01	(Ducrot et al., 2016) [32]	GDA, 5-color Nutrition Label (5-CNL), Multiple Traffic Lights (MTL), Green Tick	Numerical, color-coded, positive logos, graded indicators	Among the 18–30-year-old subgroup, positive logos were most effective in helping consumers choose healthier food products.
14	(Hodgkins et al., 2015) [44]	GDA, TL, MTL, Health logo	Numerical, color-coded, positive logos	All four FoP label formats performed similarly and all guided healthier food choices.

**Table 8 nutrients-16-02819-t008:** Key findings of graded indicators category front-of-package label studies.

Study ID	(Author, Year)	Label Formats	Type Of Fop Labels	Key Finding about Numerical Labels
01	(Ducrot et al., 2016) [32]	GDA, 5-color Nutrition Label (5-CNL), Multiple Traffic Lights (MTL), Green Tick	Numerical, color-coded, positive logos, graded indicators	Results were similar for sociodemographic and economic subgroups. The graded summary label type was the most effective, and it was significantly better than the simple summary and nutrient-specific labels.
03	(Julia et al., 2017) [34]	Nutri-Score, MTL, SENS, RIL	Graded indicators, color-coded, numerical	Nutri-Score was the most popular FoP label in most subgroups, but it was second only to MTL in the 18–30-year-old subgroup.
04	(Billich et al., 2018) [35]	Warning label, HSR	Warning labels, graded indicators	HSR labeling reduced purchase intentions for sugar-sweetened beverages, and although we were unable to determine whether participants changed their drink choices, respondents exposed to HSR labeling were more likely to choose drinks with a higher HSR.
05	(Egnell et al., 2019) [36]	Nutri-Score, RIL	Graded indicators, numerical	The Nutri-Score label improves the overall nutritional quality of a student’s shopping cart by reducing the calorie, saturated fatty acid, sodium, fiber and protein content of the cart and increasing the fruit and vegetable content.
06	(Egnell et al., 2020) [37]	MTL, RIL, Warning label, Nutri-Score, HSR	Color-coded, numerical, warning labels, graded indicators	Nutri-Score works well as a program to help research participants choose healthier diets and more precisely recognize variations in the nutritional value of different product categories. In the study, the effectiveness of HSR, another format for graded indicators label, is surpassed only by Nutri-Score.
07	(Egnell et al., 2021) [38]	Nutri-Score	Graded indicators	Nutri-Score reduces shopping cart calories and SFA (dietary score based on the food nutritional analysis method) to improve purchase intents’ nutritional quality. According to current studies, this FoP label reduces the purchasing of ultra-processed, low-nutrition foods.
08	(Egnell et al., 2018) [16]	HSR, MTL, RIL, Waring label	Color-coded, numerical, warning labels, graded indicators	The Nutri-Score summary label showed superior efficacy compared to the other four styles of the FoP labels. Consumers can comprehend the summary indications more readily, hence reducing potential uncertainty regarding the interpretation of nutritional words.
09	(Fialon et al., 2020) [39]	RI, MTL, Warning label, Nutri-Score, HSR	Color-coded, numerical, warning labels, graded indicators	The majority of participants did not alter their food preferences in either the absence of labeling or the presence of FoP labels. Furthermore, all five styles of FoP labels demonstrated comparable effectiveness in influencing the nutritional quality of food choices, both positively and negatively.
10	(Fuchs et al., 2022) [40]	Nutri-Score	Graded indicators	The use of Nutri-Score labeling has a substantial influence on the levels of saturated fat, sugar, and harmful sugar found in items chosen by consumers. The Nutri-Score system can assist consumers in enhancing the nutritional value of their food selections.
12	(Singh et al., 2022) [42]	Warning label, TLL, GDA, HSR	Warning labels, graded indicators, numerical, color-coded	This experiment found that HSR performed consistently worse than warning labels in many outcomes, including identifying high-content products and reducing purchase intentions.

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
