# Peer review of "Impact of Five Types of Front-of-Package Nutrition Labels on Consumer Behavior among Young Adults: A Systematic Review"

_nutrients, 2024, doi:10.3390/nu16172819_

Round 1

Reviewer 1 Report

Comments and Suggestions for Authors

This distinguished systematic review examines the effect of various types of food labels. The review is comprehensive and structured, original and significant.

I only have one remark: the Discussion should state that both food labels and their categories are not universal (e.g. the EFSA recognizes only four categories), which could have an effect on study outcomes.

Author Response

Comment 1: The discussion should state that both food labels and their categories are not universal (e.g., the EFSA recognizes only four categories), which could have an effect on study outcomes.

Response 1: Thanks for the advice! I have added this point in the research limitations section of the discussion chapter, making it clear that food labelling and its classification are not universally applicable, which may have an impact on the findings [line number: 395–399]. Thank you very much for pointing out this important issue, and your feedback has been very helpful in refining this section.

Finally, thank you again for your suggestions on my article.

Reviewer 2 Report

Comments and Suggestions for Authors

Dear authors,

The abstract is very well written and presents the context of the study, the aim, the methodology of the review article, the results obtained and the recommendations for future studies that could be carried out following this review, both quantitative and qualitative. 

The introduction is well written and dense, but it needs some improvement. Reading this part of the article, it is not clear what the exact research question is or what the focus of the review is. The authors should be more explicit about the controversial issue in the field. They present results from previous studies in the literature, but do not explain why these results are 'problematic' or significant. The authors discuss various aspects of FoP labelling, but do not specify the exact research question addressed in the article. In addition, the introduction contains many detailed descriptions and classifications of different labels, but sometimes these details are redundant and perhaps overly detailed. In addition, the introduction does not explicitly state the unique contribution that this review aims to make. The authors mention that the article focuses on the younger generation, but do not clarify why this aspect is important. They do not justify why they chose the younger generation and how this review differs from the existing literature. Although the introduction mentions 'young people' towards the end, most of the text does not consistently focus on this age group. This inconsistency makes the specific purpose of the review unclear. Another weakness of the introduction is that it lacks a critical tone. Simply presenting the types and examples of FoP labels without any critical examination does not effectively engage the reader. To improve the introduction, it should start with a broader context, mention the issue or controversy, explain the focus of the review and briefly outline the approach. It should avoid unnecessary detail and focus on establishing the relevance and importance of the topic of the review.

The methodological section is very well written. The authors clearly define criteria for the selection of studies. This thoroughness helps to ensure that the review includes relevant, high quality studies focusing on FoP nutrition labelling and its impact on young consumers' behaviour. The inclusion of studies specifically targeting the 18-30 age group, including students, provides a precise demographic focus, increasing the relevance and applicability of the findings to this population. The strategy of searching for studies across multiple databases (Web of Science, Scopus and EBSCO) using a comprehensive set of keywords is highly appropriate. The authors carry out a rigorous selection process of the identified studies and evaluate them according to eight criteria. The choice of narrative synthesis is also appropriate given the diversity of studies and their designs. A weakness of the methodology is the exclusion of studies in languages other than English, which neglects important research conducted in other languages. This limitation may affect the comprehensiveness and overall applicability of the review findings.

The results are very well presented. The study finally included 14 papers published between 2015 and 2023, covering several countries (France, Australia, New Zealand, Switzerland, Italy, Uruguay, Jamaica, India and cross-national studies) and different study designs. The quantitative studies included in the results and the large sample sizes of these studies provide very robust data. These elements add depth to the findings. The study also evaluated five different types of FoP labels: numeric, coloured, warning, positive logos and graded indicators. This comprehensive categorisation allows a detailed analysis of different labelling strategies and their impact on consumer behaviour. In presenting the results, the authors analysed different age subgroups in the 18-35 age group, and the consistency of the results across studies adds credibility. For example, the effectiveness of coloured labels and graded indicators (e.g. Nutri-Score) was frequently reported, suggesting a generalisable trend in consumer behaviour towards these types of labels. As a suggestion for improving the results, the authors should present the relevance of focusing on young consumers (e.g. students, young adults), as this population is often a key target for public health interventions and marketing strategies.

The conclusions of the study are superficial compared to the whole article. They should be improved.

Author Response

Comment 1: Reading this part of the article, it is not clear what the exact research question is or what the focus of the review is. The authors should be more explicit about the controversial issue in the field. They present results from previous studies in the literature, but do not explain why these results are 'problematic' or significant. The authors discuss various aspects of FoP labeling but do not specify the exact research question addressed in the article.

Response 1: We agree. Therefore, we have restructured the introduction to include the research question in the [fifth paragraph]. It is also clearly pointed out that the previous studies did not review the 5 categories of FoP labels in detail in the summary of the effectiveness of different categories of FoP labels but summarized them with interpretative labels, reductive labels or summary labels, and nutrient-specific labels.

Comment 2: The introduction contains many detailed descriptions and classifications of different labels, but sometimes these details are redundant and perhaps overly detailed.

Response 1: We agree with this recommendation. Therefore, we have removed some examples and introductions in the FoP label classification section.

Comment 3: The introduction does not explicitly state the unique contribution that this review aims to make.

Response 3: We added the unique contribution of this paper in the [fifth paragraph]: The insights gained from this study are the basis for a deeper understanding of FoP nutrition labels, which will provide a review of the gaps in the performance of FoP labels in young subgroups and will be important for scholars who are interested in and have a preliminary understanding of research in this field, as well as governments and institutions that are trying to develop new FoP label design models.

Comment 4: The authors mention that the article focuses on the younger generation, but do not clarify why this aspect is important. They do not justify why they chose the younger generation and how this review differs from the existing literature. Although the introduction mentions 'young people' towards the end, most of the text does not consistently focus on this age group.

Response 4: We agree. Therefore, we mentioned at the beginning of the introduction that there has been a sharp rise in the prevalence of overweight and obesity, and this upward trend is also seen in the younger population. and explains the harms of this trend [line numbers: 28-32]. In the [fifth paragraph], we also describe the gaps in summarizing and reviewing the effects of FoP labels on the consumer behavior of younger people and the importance of studying FoP labels for different population subgroups.

Comment 5: Another weakness of the introduction is that it lacks a critical tone. Simply presenting the types and examples of FoP labels without any critical examination does not effectively engage the reader.

Response 5: We intentionally added critical tone. For example, Ikonen et al. (2019) pointed out that some types of FoP labeling systems have been criticized for their differences in leading consumers to make healthier choices. However, the categories of FoP labels summarized in this article only include interpretative labels and reductive labels, or summary labels and nutrient-specific labels, and do not mention the specific 5 label categories in detail. This can lead to some confusion for some scholars who are new to the field [line number: 99-105].

Comment 6: A weakness of the methodology is the exclusion of studies in languages other than English, which neglects important research conducted in other languages. This limitation may affect the comprehensiveness and overall applicability of the review findings.

Response 6: Thank you for your suggestion, but it is true that due to the limited capacity of our team, we can only include English articles for the time being. I admit that it leads to certain limitations.

Comment 7: As a suggestion for improving the results, the authors should present the relevance of focusing on young consumers (e.g., students, young adults), as this population is often a key target for public health interventions and marketing strategies.

Response 7: Thank you for your advice. We describe the relevance of the included studies to young consumers in the results of the study quality assessment [line tone: 229-235].

Comment 8: The conclusions of the study are superficial compared to the whole article. They should be improved.

Response 8: Thank you for your feedback! I have improved the conclusion section, optimized the summary of the main research findings, added some limitations of the study and suggestions for future research, as well as the practical application impact of the research results. I hope that the current conclusion can more fully reflect the depth and value of the entire article [line: 428-449].

Finally, thank you again for your suggestions on my article.

Reviewer 3 Report

Comments and Suggestions for Authors

From an academic review standpoint, while the manuscript provides a significant contribution to Front-of-package Nutrition Labels, several areas could be enhanced to improve the overall quality and impact of the research. Below are some deficiencies and suggestions for improvement:

1.The introduction jumps between topics without smooth transition. It starts by introducing nutritional package labelling, then quickly shifts to the World Health Organization's recommendations, then to specific examples, and finally to typologies of FoP labels without clear connections between these segments.

2.Some information is repeatedly or redundantly stated. For instance, the purpose of FoP labels to help consumers make healthier choices is mentioned multiple times (lines 33-36 and again in lines 35-41).

3.The paragraph structure is non-ideal. Some paragraphs are overly long, making them difficult to digest, whereas others are too short, disrupting the flow of reading.

4.The research objective is buried under extensive background information, which may make it less prominent. A clearer and more concise statement of the research aim would help guide the readers.

5.There are some ambiguous statements (e.g., "various labelling systems pursue this objective through different methods, which may result in divergent outcomes" (lines 40-42)). This needs to be clarified to enhance our understanding.

6.Initial screening by a single reviewer (G) may have introduced a bias. Industry standards usually recommend at least two independent reviewers to minimize subjective bias.Terms like "likely included" and "maybe included" lack clear operational definitions, which can introduce subjectivity into the screening process.

7.No clear definitions or criteria have been established for “the reliability and validity of these measurements,” which could lead to subjective assessments. It is unclear whether any measures such as funnel plot analysis were used to assess publication bias, which is critical when interpreting narrative synthesis.

8.The coding strategy must be thoroughly explained with an emphasis on ensuring consistency among different reviewers.

9.The Results section should include a more critical analysis and synthesis of the findings. It primarily presents findings descriptively rather than analytically comparing the methodologies, contexts, and outcomes of different studies. There is a need to identify patterns, commonalities, and discrepancies across studies more explicitly and to discuss their potential causes (e.g., cultural differences, types of food products, and different study populations). 10.The discussion heavily emphasizes the role of color in label effectiveness, but overlooks other potential influential factors, such as label placement, simplicity of information, and cultural influences. This narrow focus could be expanded to provide a more comprehensive understanding.

11.There is a lack of integrated and critical analyses of diverse studies. The review mentions specific studies (e.g., Julia et al., Egnell et al.) but does not compare or contrast these findings in detail with other research, nor does it critically analyze the methodologies and sample sizes of the cited studies.

12.Although the discussion acknowledges that few studies have examined these categories, it does not adequately address why these gaps exist or their implications. This would benefit from a discussion of the potential biases or limitations in the current body of research.

13.The mention of limitations (e.g., language restrictions and lack of backward and forward reference searches) is brief and can be expanded to discuss how these limitations might have specifically affected the study results. Additionally, there was no mention of a potential publication bias or heterogeneity among the included studies.

14.While this section includes recommendations for future research, it lacks detailed actionable steps. For example, it suggests studying more comprehensive design elements but does not specify which elements should be included or how these studies should be designed.

15.The statement that “the majority of FoP nutrition labelling schemes are effective with younger consumers” is an overgeneralization and lacks evidence-based support. The wording suggests conclusive effectiveness in that the mixed results of some categories (e.g., warning labels) are not justified.

16.Terms such as ‘most effective’ and ‘little efficacy’ are vague. The conclusions would benefit from quantitative data or a clearer explanation of the meaning of these terms in the context of the reviewed studies.

17.The conclusions do not acknowledge or address any potential limitations of the systematic review, such as bias in study selection, variability in study design, or the diversity of the populations studied.

18.There is no discussion on how these findings can be practically applied, or what the implications might be for policymakers, food manufacturers, or health educators aiming to promote healthier eating behaviors among young adults.

Author Response

Comment 1: The introduction jumps between topics without smooth transition. It starts by introducing nutritional package labelling, then quickly shifts to the World Health Organization's recommendations, then to specific examples, and finally to typologies of FoP labels without clear connections between these segments.

Response 1: Thank you for your comments, and we have once again structured the framework for the introduction. First, we will talk about the issue that leads to the necessity of the FoP labels strategy, and the definition and purpose of FoP labelling. The benefits of the FoP label and the stakeholders were then mentioned. The next two paragraphs describe the classification of FoP labels. This is followed by an explanation of the consumer behavior that is the focus of this article. Finally, we talk about some of the challenges and gaps of the FoP label, as well as the research question, research purpose, and contribution of this paper.

Comment 2: Some information is repeatedly or redundantly stated. For instance, the purpose of FoP labels to help consumers make healthier choices is mentioned multiple times (lines 33-36 and again in lines 35-41).

Response 2: Thanks for the reminder. After checking, the duplicate parts have been removed.

Comment 3: The paragraph structure is non-ideal. Some paragraphs are overly long, making them difficult to digest, whereas others are too short, disrupting the flow of reading.

Response 3: Thank you for your suggestion. Some passages have been reduced and segmented to ensure ease of digestion and fluency for readers. For example, the FoP tag classification in the introduction (second and third paragraphs).

Comment 4: The research objective is buried under extensive background information, which may make it less prominent. A clearer and more concise statement of the research aim would help guide the readers.

Response 4: In the fifth paragraph of this article, we add some critical statements to better describe some gaps in the FoP labels, and then logically arrive at the research questions and objectives of this paper to guide the reader more clearly.

Comment 5: There are some ambiguous statements (e.g., "various labelling systems pursue this objective through different methods, which may result in divergent outcomes" (lines 40-42)). This needs to be clarified to enhance our understanding.

Response 5: Thank you for your comments. We have changed this statement and checked elsewhere. Let's briefly explain the meaning of your example sentence: Some types of FoP labeling systems have been criticized for their differences in leading consumers to make healthier choices [line number: 99-101].

Comment 6: Initial screening by a single reviewer (G) may have introduced a bias. Industry standards usually recommend at least two independent reviewers to minimize subjective bias. Terms like "likely included" and "maybe included" lack clear operational definitions, which can introduce subjectivity into the screening process.

Response 6: We agree with this comment. Therefore, we (G&N) re-conducted the first round of screening and identified 86 articles that were marked for a second round of screening. Fortunately, these 14 articles were included in the review. Cohen's Kappa measured 0.83, and the consistency was almost perfect!

Comment 7: No clear definitions or criteria have been established for “the reliability and validity of these measurements,” which could lead to subjective assessments. It is unclear whether any measures such as funnel plot analysis were used to assess publication bias, which is critical when interpreting narrative synthesis.

Response 7: Thank you for your comment. I've learned that almost a lot of meta-analyses use funnel plot analysis to assess publication bias. I didn't find a guideline that instructed me to use the evaluation method to assess the publication bias of the narrative synthesis. Do you have any relevant literature to share with me? Of course, I think this is a very good suggestion, and I put this in the future research recommendations of this article [line: 437-443].

Comment 8: The coding strategy must be thoroughly explained with an emphasis on ensuring consistency among different reviewers.

Response 8: Coding strategies have been added to the article. Reviewer G independently performs the initial coding, and reviewer N then verifies all data extractions [line: 185-186].

Comment 9: The Results section should include a more critical analysis and synthesis of the findings. It primarily presents findings descriptively rather than analytically comparing the methodologies, contexts, and outcomes of different studies. There is a need to identify patterns, commonalities, and discrepancies across studies more explicitly and to discuss their potential causes (e.g., cultural differences, types of food products, and different study populations).

Response 9:  Thank you for your detailed feedback, I very much understand your suggestion for more in-depth analysis and synthesis. Indeed, analyzing the differences and commonalities between different studies can provide more insights into our research. However, I would like to explain that I have chosen a descriptive approach in the results section, mainly to ensure that all readers, regardless of their background, can clearly understand the main findings of the study. At the same time, I think each study has its own unique context and methodology, and direct comparisons may obscure these uniquenesses. Therefore, I chose to explore these differences and similarities further in the discussion section and avoid introducing too much analytical detail too early in the results section. However, I would consider adding some preliminary comparisons and analyses to the results section to better guide the reader through the relationship between the studies. Thank you again for your valuable advice, and I will continue to work on refining this section. 

Comment 10: The discussion heavily emphasizes the role of color in label effectiveness, but overlooks other potential influential factors, such as label placement, simplicity of information, and cultural influences. This narrow focus could be expanded to provide a more comprehensive understanding.

Response 10: Thank you for your meticulous feedback! I've taken your suggestion into account and added a discussion of other design elements to the discussion section, such as the size and position of labels, graphics, and icons, and the impact of the simplicity of information [line: 356-358]. I believe these additions will help provide a more comprehensive understanding.

Comment 11: There is a lack of integrated and critical analyses of diverse studies. The review mentions specific studies (e.g., Julia et al., Egnell et al.) but does not compare or contrast these findings in detail with other research, nor does it critically analyze the methodologies and sample sizes of the cited studies.

Response 11: Thank you very much for your suggestion. I've added some critical analysis to the results, especially in the quality analysis results [line: 221-235].

Comment 12: Although the discussion acknowledges that few studies have examined these categories, it does not adequately address why these gaps exist or their implications. This would benefit from a discussion of the potential biases or limitations in the current body of research.

Response 12: Thanks for the reminder. I have added to the study limitations in the discussion section a restriction that the classification of FoP labels is not universally applicable, for example, the European Commission proposed to recognize only four categories of FoP labels, excluding warning labels [line: 396-398]. Discussing this potential bias would really benefit the article, thank you!

Comment 13: The mention of limitations (e.g., language restrictions and lack of backward and forward reference searches) is brief and can be expanded to discuss how these limitations might have specifically affected the study results. Additionally, there was no mention of a potential publication bias or heterogeneity among the included studies.

Response 13: Thanks for the detailed suggestion! I have expanded on the discussion of language limitations and the absence of forward and backward search in the Research Limitations section of the discussion section, detailing how these limitations may affect the results of the study [line: 375-378, 386-389].

Comment 14: While this section includes recommendations for future research, it lacks detailed actionable steps. For example, it suggests studying more comprehensive design elements but does not specify which elements should be included or how these studies should be designed.

Response 14: Thanks for the suggestion! I've added some detail to the relevant sections, clarifying which design elements should be included and what research designs might be applied [line: 407-417]. I believe that these additions will make future research directions clearer and more concrete. 

Comment 15: The statement that “the majority of FoP nutrition labelling schemes are effective with younger consumers” is an overgeneralization and lacks evidence-based support. The wording suggests conclusive effectiveness in that the mixed results of some categories (e.g., warning labels) are not justified.

Response 15: Thank you for your suggestion. I have added a qualification to this general summary statement. It now reads: In general, most FoP labelling schemes are effective to some extent for young consumers, but the effectiveness varies depending on the label category and the research subjects [line: 434-436].

Comment 16: Terms such as ‘most effective’ and ‘little efficacy’ are vague. The conclusions would benefit from quantitative data or a clearer explanation of the meaning of these terms in the context of the reviewed studies.

Response 16: Thank you for your advice. I've made changes to this type of term in the article. such as in the abstract and conclusion [line: 16-20, 428-436]. 

Comment 17: The conclusions do not acknowledge or address any potential limitations of the systematic review, such as bias in study selection, variability in study design, or the diversity of the populations studied.

Response 17: Thank you for your valuable comments. I have added a discussion of the potential limitations of systematic reviews to the revised draft, and I hope that these additions will more fully reflect the rigor of the research [line: 437-443].

Comment 18: There is no discussion on how these findings can be practically applied, or what the implications might be for policymakers, food manufacturers, or health educators aiming to promote healthier eating behaviors among young adults.

Response 18: Thank you for your feedback. I have added a discussion of the practical application of these findings and their potential impact on government agencies and nutrition facilities in the revised draft [line: 444-449].

Finally, thank you again for your suggestions on my article.

Reviewer 4 Report

Comments and Suggestions for Authors

1.      There is no author information at all on the manuscript. Is there any reason why it cannot be disclosed? Therefore, I cannot clearly determine whether it is necessary to avoid review.

2.      Figure 1 is very blurry. It is recommended to readjust it. Or, please put the original image file in the appendix.

3.      There is no information about Author Contributions, Funding, Acknowledgments and Conflicts of Interest at the end of this manuscript. The above information should be supplemented.

4.      The reference citation format is incorrect and should be revised to comply with the journal's requirements.

Author Response

Comment 1: There is no author information at all on the manuscript. Is there any reason why it cannot be disclosed? Therefore, I cannot clearly determine whether it is necessary to avoid review.

Response 1: Sorry. We didn’t know it was a single-blind journal. We thought it was a double-blind journal, so we didn’t put my name and institution in the article. After your reminder, we added the  author information to the article. Thank you.

Comment 2: Figure 1 is very blurry. It is recommended to readjust it. Or, please put the original image file in the appendix.

Response 2: Thank you for your reminder, I will upload the clear pictures into the supplementary.

Comment 3: There is no information about Author Contributions, Funding, Acknowledgments and Conflicts of Interest at the end of this manuscript. The above information should be supplemented.

Response 3:Thank you for your suggestion. I have added information on author contributions, funding, acknowledgments, and conflicts of interest to the end of my manuscript.

Comment 4: The reference citation format is incorrect and should be revised to comply with the journal's requirements.

Response 4: Thank you for your feedback. I will carefully review and revise the format of the references to ensure that they meet the journal's requirements. Could you provide specific examples or instructions to help me make the corrections more accurately?

Finally, thank you again for your suggestions on my article.

Round 2

Reviewer 3 Report

Comments and Suggestions for Authors

Dear authors,

Thank you for your comprehensive response to my review comments and for addressing the suggestions provided. I have carefully reviewed the revisions and am pleased to see that my concerns have been satisfactorily addressed. I believe the manuscript has been significantly improved and is now suitable for publication.

Author Response

Dear Reviewer,

Thank you very much for your review and affirmation of our manuscript. We are very grateful for your valuable suggestions and guidance, which played an important role in improving the quality of the manuscript. We are glad to know that the revised manuscript has met your expectations and is suitable for publication.

Thank you again for your support!

Kind regards,

Guo Zhiyi

Reviewer 4 Report

Comments and Suggestions for Authors

1.      The way of marking literature references in the manuscript does not comply with common rules.

For example, LINE 194: [32,33,34,35,36,37,38,16,39,40,41,42,43,44] should be adjusted to [16,32-44].

Please adjust the rest in this way.

2.      Figure 2 is very blurry. It is recommended to readjust it. Or, please put the original image file in the appendix.

3.      The reference citation format is still incorrect and should be revised to comply with the journal's requirements. The journals cited should be presented by abbreviation.

Author Response

Dear reviewer, 

Thank you very much for your suggestions. I have made some changes and will explain them one by one below.

Comment 1:  The way of marking literature references in the manuscript does not comply with common rules.

Response 1: Thank you for your reminder! I have corrected the citation method according to the journal's rules.

Comment 2: Figure 2 is very blurry. It is recommended to readjust it. Or, please put the original image file in the appendix.

Response 2: Thanks for your suggestion! I have re-adjusted the clarity of Figure 2 and adjusted the display angle of Figure 2 to ensure a clearer image.

Comment 3: The reference citation format is still incorrect and should be revised to comply with the journal's requirements. The journals cited should be presented by abbreviation.

Response 3:Thank you for your reminder! I have corrected the reference format according to the journal's requirements.

Thank you again!

Kind regards,

Guo Zhiyi